# Development of Preliminary Curved Bamboo Member Design Guidelines through Finite Element Analysis

**Faham Tahmasebinia [1,*]**, **Rory McDougall [2]**, **Samad Sepasgozar [3]**, **Emma Abberton [2]**,
**Gi Houn Joung [2]**, **Maria Paula Joya [2]**, **Saleh Mohammad Ebrahimzadeh Sepasgozar [4]** and
**Fernando Alonso Marroquin [2]**

[1]   School of Minerals and Energy Resources Engineering, University of New South Wales,
     Sydney 2052, Australia

[2]   School of Civil Engineering, The University of Sydney, Sydney, NSW 2006, Australia;
     rmcd8763@uni.sydney.edu.au (R.M.); eabb5455@uni.sydney.edu.au (E.A.);
     gjou5781@uni.sydney.edu.au (G.H.J.); mjoy5213@uni.sydney.edu.au (M.P.J.);
     fernando.alonso@sydney.edu.au (F.A.M.)

[3]   Faculty of Built Environment, The University of New South Wales, Sydney, NSW 2052, Australia;
     Samad.sepasgozar@gmail.com

[4]   Babol Noshirvani University of Technology, Babol 47148, Iran; s.sepasgozar@yahoo.com

*   Correspondence: F.tahmasebinia@unsw.edu.au

**Abstract:** Bamboo is emerging as a lightweight, versatile and renewable material that is projected to realise new methods of construction. There is a growing demand for using bamboo in different regions across the world. However, there are no specific design standards or guidelines that capitalise on the unique circular hollow cross section and internal nodal support of bamboo. Furthermore, there has been no strict analysis into how the curvature of bamboo members can help to better distribute stress through a structure. Bamboo is known as a more environmentally sustainable material than standard timber; however, due to the naturally occurring diaphragm nodal structure, it is less orthotropic in mechanical behaviour, with more out of plane stiffness than timber. To address this issue, this paper presents finite element analysis of three varying bamboo structures, each featuring a varying member curvature and lateral support system. In this paper, a preliminary set of design guidelines have been proposed for bamboo members, maximising the performance of its inherent mechanical properties. These guidelines propose the use of thick, low diameter bamboo members in axial loading and thin, high diameter members in flexural situations. Where available, the preliminary guidelines introduce the importance of curved bamboo members to uniformly distribute forces and crossing arched members to eliminate the need for lateral support systems, thereby reducing the total material required for construction. Finally, this report presents some novel consideration of the out-of-plane buckling of curved bamboo members–although for this purpose it does not consider the effect of the diaphragm components of bamboo, a source of future research and more comprehensive design guidelines.

**Keywords:** Bamboo; Curved Beams; Advanced Analysis; Finite Element Analysis; Design Guidelines

## 1. Introduction

Building construction is known as one of the major contributors to global carbon emission [1–5]. For example, Kumanayake and Luo [4] recently reported that the building sector consumes over 50% of raw materials for construction purposes in Sri Lanka. Sepasgozar and Blair [6] also reported that

emissions in Australia form a high contribution to the total nitrogen oxide (NOx) and particulate matter (PM) emissions at the national level due to various construction activities, including handling materials and diesel engines and equipment utilised in the construction phase. However, most studies have tended to develop formulas for estimating the emission or estimate the embodied carbon in building materials. There is a growing literature offering new materials with less footprint which are mainly composite [7], and some of them still include concrete or steel as a part of the material [8]. However, the literature offering techniques and technologies to facilitate the utilisation of natural renewable materials such as bamboo is scarce. The aim of this paper is to present how bamboo can be utilised as the main material of buildings by designing a series of bamboo grid shell structures and develop a series of design guidelines using this material. This will be done through a numerical investigation of the strength and serviceability limits via means of finite element analysis. Three structures will be designed using varying curved member designs and lateral support systems, and compared with respect to the distribution of axial, fibre, bending and combined action stresses. This global analysis will be performed to examine the general behaviour of the structure under a combination of dead, live and wind loads. To achieve this, all three preliminary designs were modelled in AutoCAD [9] and imported into Strand7 [10] for property assignment, loading and analysis.

The innovation of this study comes principally from the choice of material and particularly the incorporation of curved member design. Historically, bamboo has been used for thousands of years in the construction of houses and bridges in various tropical regions [5,11]. However, due to the lack of consistency in its geometry and properties, bamboo has been considered an unconventional material for modern constructions and remains associated with rural and artisanal constructions. Moreover, using bamboo as a primarily straight member, due to its method of growth, means that there has been limited exploration of using bamboo in a curved roof context. A review of the recent literature has demonstrated a renewed interest for bamboo as a sustainable substitute for more conventional materials and purported [12–16] excellent structural performance in curvature. Bamboo is considered a renewable material given its abundance and the remarkable speed at which it grows, and presents a relatively untouched source of sustainable design. This natural material is also known for its strength, flexibility, low weight and reported favourable action when curved, making it ideal for roofing structures. Thus, combining it with glass in a grid shell design allows for the creation of modern looking, aesthetically pleasing, low-cost and sustainable structures.

Although there is a growing interest in the community to use bamboo as a construction material, its mechanical properties have only had limited analysis, and there are no established design standards and guidelines for its applications and implementation. The goal of this research project is to shed some light on the feasibility of bamboo as a construction material through the establishment of preliminary design guidelines. With these preliminary design guidelines, the aim is to encourage the further study of bamboo and increase its use in modern architectural and civil engineering projects.

## 2. Background Review

### 2.1. Bamboo Material Properties and Failure Mechanisms

Bamboo is a renewable sustainable material and its mechanical properties has some similarities to timber. It is a cost-effective and recommended material for ecology [17–19]. Bamboo was selected as the main structural material for this research. The increased interest in bamboo as a structural material is driven from the perspective of sustainability [20]. The material is fast growing and renewable, with strong structural properties. The material is not commonly used due to the non-homogenous cross-sectional shape and common irregularities [21]. There are numerous characteristics which impact the structural properties of bamboo, including climate, topography, soil, altitude, cutting, treatment, age, stem and humidity [22]. The variance of bamboo properties creates uncertainty in the applicable properties for analysis of a bamboo structure. The range of properties required in the analysis is detailed in Table 1.

**Table 1.** Common ranges of bamboo's material properties.

| Property | Value |
|---|---|
| Elastic Modulus (E) [23] | 5000–25,000 MPa |
| Poisson's Ratio [24] | 0.26–0.52 |
| Density (p) [25] | 300–400 kg/m$^3$ |
| Diameter (D) [26] | 70–170 mm |
| Thickness (t) | 8–18 mm |

From understanding these properties, experimentation has been conducted into the limiting capacity of bamboo under different failure mechanisms. Bamboo is susceptible to cross-sectional flattening causing global failure, by way of tangential tension perpendicular to the grain [27]. This failure type is a consequence of exceeding of failure stress of specific loading mechanisms, which results in the local failure of the bamboo. The local failure stress of each loading mechanism can be found in Table 2.

**Table 2.** Failure stresses of bamboo by varying mechanisms.

| Property | Value |
|---|---|
| Tension [28] | 115 MPa |
| Compression [28] | 108 MPa |
| Bending [29] | 97.1 MPa |
| Shear [28] | 29.1 MPa |
| Fibre [30] | 43.5 MPa |

The exceedance of any of the values in Table 2 results in local failure of the bamboo by that mechanism, in turn causing flattening of the bamboo member and global failure. These values will be considered throughout the structural analysis of the proposed structures in this report.

*2.2. Bamboo Compared to Sawn Timber*

The use of bamboo in construction presents some challenges in design, due to its flammable nature and material defects. However, due to its light weight and flexibility, the material is proven to be highly resistant to earthquakes and structurally strong, despite inconsistencies [22]. The mechanical properties of bamboo vary along the length of the stems and radially in diameter, making it an anisotropic material and making it varied from usual timber applications. A comparison of material properties for specific species of timber and bamboo are compared in Table 3 to demonstrate the applicability of bamboo as a structural material. Table 3 indicates the relatively high capacity of bamboo compared to other sawn timber types relative to its density. This is a result of the natural structure of bamboo providing additional support perpendicular to the grain by way of diaphragms [31].

**Table 3.** Material properties for structural timber comparison [32].

| Material | Density $\rho$ kg/m$^3$ | Compression $f_c$ MPa | Tension $f_t$ MPa | Shear $\tau$ MPa | Flexural $f_b$ MPa |
|---|---|---|---|---|---|
| Bamboo | 425 | 108 | 115 | 29 | 97 |
| Sitka spruce | 383 | 36 | 59 | 9 | 67 |
| Douglas-fir | 520 | 57 | 49 | 11 | 68 |

The diaphragm provides transverse support to bamboo that is otherwise missing from mass timber. In the diaphragms, bamboo fibres are not longitudinal to the plane of the bamboo—instead, they are perpendicular, preventing failure by making the tubular structure stiffer out of plane [31]. This provides a higher capacity than mass timber in planes perpendicular to the grain, as the orthotropic

structure of timber lacks any considerable capacity not longitudinal to the grain. As an additional consequence, bamboo has an inherently greater flexural capacity and resistance to buckling.

### 2.3. Design of Grid Shells

Bamboo grid shells formed the design inspiration for this structural analysis and design research. A grid shell is composed of individual members connected to create a structural form which is light and strong, through a curved form-active structure [33]. This type of structural form reduces the cost and complexity of frameworks associated with alternative shell structures [33]. The designs of the bamboo structural systems are combined with thin glass as a cladding material between the bamboo members [34]. This thin glass is cold bent on site and integrated with the structural form to overcome the inconsistencies in the bamboo structure [21]. This material combination also allows for a waterproof sheltered structure with natural ventilation from the side openings [35].

### 2.4. Curvature

The method of curvature is directly relatable to bamboo's material properties, specifically with cold bending [36]. Cold bending includes bundling—the act of splitting bamboo members and binding back together in a curved shape—and lashing with string into a desired shape. This process results in a decrease in compressive and tensile strength, respectively. Hot bending, however, such as immersion in hot water for an extended period or exposure to small amounts of fire, experimentally have shown no decrease in mechanical properties after curvature. Therefore, if the bamboo is curved by way of hot bending, there is no reduction of capacity when compared to straight bamboo members. In fact, curvature introduces greater flexure into the bamboo members, which, as a result of the diaphragm structure, better utilises the material properties of bamboo compared to straight members [37].

When considering curved members subjected to compressive loading, as is the case with dead loads, buckling analysis of the members must be performed. Timoshenko's theory of elastic stability presents some limited analysis of curved member buckling; however, it inherently includes elastic behaviour approximations that will overestimate capacity [38]. There has been further progression of this theory through the application of the Rayleigh–Ritz energy method to provide a more accurate theoretical result [39]. This form of analysis, however, requires rigorous mathematical calculation, whilst failing to consider any further sources of restraint, such as lateral support mechanisms or cladding. However, finite element analysis has been shown to present equivalent values to this form of calculation, eliminating the need for more than one type of analysis [40].

### 2.5. Connection Type

The connection between bamboo members in a grid shell is required to prevent the sliding or torqueing of members, enabling a fixed joint [33]. This connection also allows the transfer of forces between members [41]. The crossing of members provides an inherent lateral support system, in effect distributing loads more effectively through the structure than conventional lateral restraint, such as bracing. The round hollow section of bamboo prevents the use of similar joints as other wooden structures. There are two optional solutions for the connection between bamboo members, including binding and bolted connections [42].

### 2.6. Design Standards and Curved Structures

There is limited specific information with regard to curved roofs available in design standards. Curved structures provide less localisation of stress when compared to more conventional rectangular building forms, that are susceptible to stress concentration at pointed corners. As curvature radius increases, stress and strain distributions become more uniform, utilising the capacity of the entire member and therefore producing a more efficient design [43].

Furthermore, curved structures are more aerodynamically efficient shapes, as wind must move faster over the curved shape and therefore have a lower pressure [44]. Eurocode 5 [45] is the only wind

design standard that has specific pressure coefficients for curved roof forms; however, it presents a simplified calculation mechanism for these pressures. Eurocode 5 [45] assumes that a curved roof can be approximated as a windward region, top region and leeward region, with a constant pressure throughout each. This approximation models a curved roof as a scaled rectangular structure with three effective surfaces, as opposed to considering the change in curvature throughout. Despite this, the approximation serves as a reasonable representation of wind distribution within the scope of this analysis [44].

## 3. Numerical Modelling Strategy

The initial designs of the structural system are provided below—three options are provided which will be analysed and compared throughout this report. The structural system in all design options consists of curved bamboo members positioned to form a grid like structure. Each system utilises arched members to resist both vertical loads, with bracing in Design 2 and arched member crossover in Designs 1 and 3 to resist lateral loads. In all designs, the bamboo structural members support thin glass roofing panels. As detailed in Tables 2 and 3, bamboo is strong in both compressive and tensile strength making it a valuable material for resisting buckling. This material combination with glass compensates for the shortcomings of bamboo regarding the discussed lack of consistency in size and properties.

All bamboo members were designed using beam elements with the same cross-sectional geometry shown in Figure 1 below. Throughout the modelling process, the members were initially modelled as 70 mm diameter, 12 mm thick circular hollow sections. Therefore, this did not account for the potential variation of diameter inherent in bamboo, nor the diaphragm sections of bamboo. The diaphragm component of bamboo provides a considerable increase in capacity of bamboo compared to a conventional circular hollow section, so the omission of this component presents a conservative analysis to account for diameter variation.

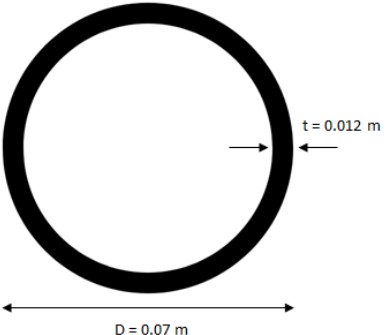

**Figure 1.** Beam element cross-sectional geometry used to model bamboo members.

The values input into Strand7 for the sectional properties are provided in the below Table 4, which is based on the researched conducted by [28,30] (see Table 1). These values were applied as beam properties following the initial design import and modification into Strand7.

**Table 4.** Selected material properties of bamboo utilised in Strand7 models.

| Property | Value |
| --- | --- |
| Elastic Modulus (E) | $1.5 \times 10^{10}$ Pa |
| Poisson's Ratio | 0.46 |
| Density (p) | 425.0 kg/m$^3$ |
| Diameter (D) | 0.07 m |
| Thickness (t) | 0.012 m |

The design of joints remained consistent across the three designs and was through the fixed connections.

### 3.1. Structural System Design 1

The below sketches in Figure 2 and Table 5 show in detail the proposed Design 1 for the structural system. Design 1 utilises circular arches that cross over in order to introduce lateral support to the structure. This crossover allows for transmission of stresses into neighbouring members, allowing for more effective load distribution. There are additional horizontal lateral support bamboo beams; however, the amount required is reduced by the crossed arch lateral support. Furthermore, the vertical loads are resisted by the arch structures, transferring the forces into the fixed base connections by way of thrust.

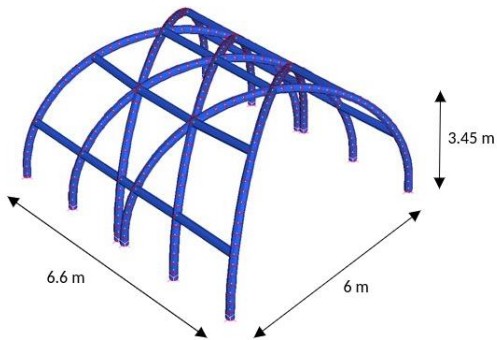

**Figure 2.** Structural system Design 1 isometric view, showing dimensions in three orthogonal directions.

**Table 5.** Design 1 structural system sketches.

| Plan | Front Elevation | Side Elevation |
| --- | --- | --- |
| | | |

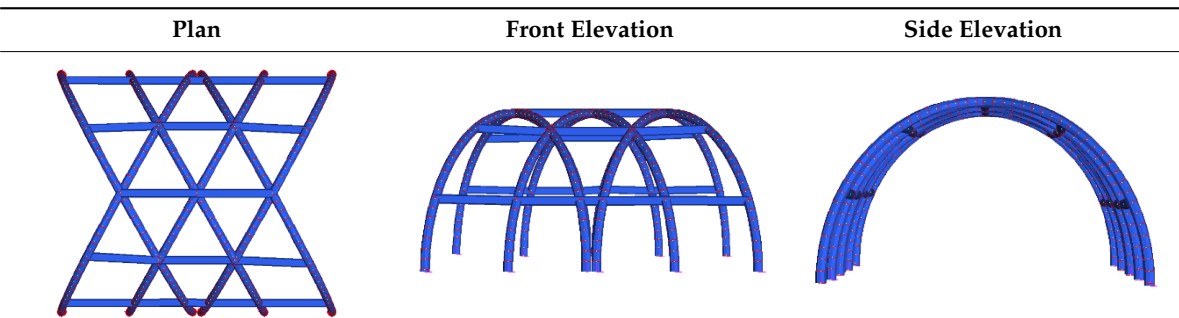

The tributary width of the structure is at a maximum of 1.7 m at the base of the structure, decreasing with proximity to the top of the design. The glass roofing panels are only affixed to the bamboo structure at points of arch crossover, to avoid any further restraint mechanisms that will affect buckling analysis. The use of diagonal arches results in a larger radius of curvature than if the arches were perpendicular to the plane of the structure. This attempts to distribute forces more uniformly through each arch, as well as increasing usable space by increasing the headroom at the arch base.

### 3.2. Structural System Design 2

The below sketches in Figure 3 and Table 6 show in detail the proposed Design 2 for the structural system. Design 2 utilises a more conventional arch design, in a typical barrel vault formation. There is no crossover of arches in this design, meaning that the transfer of lateral force is only via the horizontal bamboo members. As a result, the absence of direct arch to arch transmission of lateral force requires a considerable amount of horizontal bracing, increasing the amount of bamboo material required for this design. The arches resist vertical loading and distribute forces into the fixed base connections of the structure by thrust. The tributary width of the arches is a constant 1.65 m throughout, with the

self-weight of glass loading directly onto the arches. Furthermore, the glass is attached at the locations of horizontal to arched bamboo connections to avoid the introduction of further restraint that will affect buckling analysis. By use of arches directly perpendicular to the plane of the structure, force distribution uniformity will be limited, as opposed to using larger radius of curvature beams. Despite this, the relatively large radius of curvature will allow for fair uniformity of stress distribution, albeit with reduced usable space, due to the reduced headroom near the arch base.

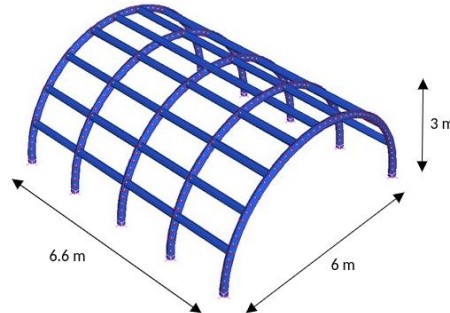

**Figure 3.** Structural system Design 2 isometric view, showing dimensions in three orthogonal directions.

**Table 6.** Design 2 structural system sketches.

| Plan | Front Elevation | Side Elevation |
| --- | --- | --- |
| 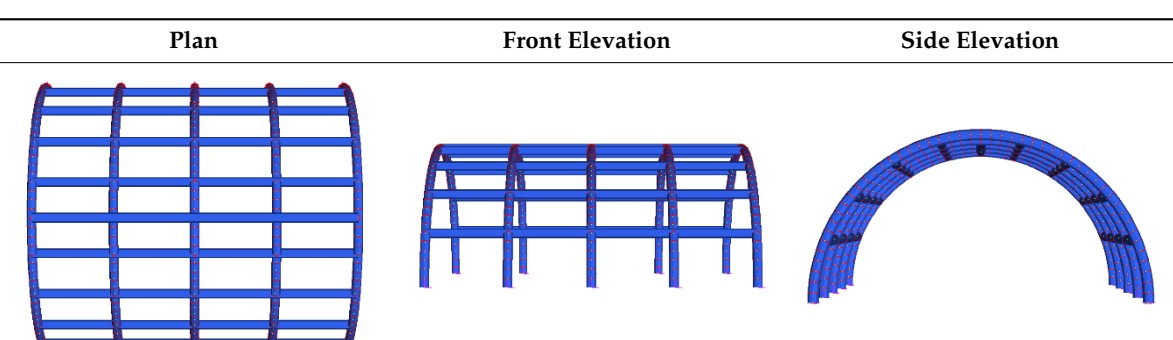 | | |

## 3.3. Structural System Design 3

The below sketches in Figure 4 and Table 7 show in detail the proposed Design 3 for the structural system. Design 3 utilises a combination of elements from Designs 1 and 2. It incorporates both crossed-over and non-crossed arches, in order to provide additional lateral and vertical support respectively. There is minimal use of lateral support due to the large radius of curvature of the crossed over arches, providing considerable member capacity and uniform lateral stress distribution. In all cases, forces are transferred down arches to the base connections by way of thrust. The minimalist design utilised in this system results in a large tributary width for the diagonal arched members of 3.3 m. Furthermore, there must be restraint of the glass panels along the arches at locations where there is no previous restraint, in order to avoid points of concentrated glass self-weight loading. This added restraint will decrease the effective length of the member for out-of-plane buckling analysis in practice; however, for analysis purposes, this will be ignored. Due to the use of arches perpendicular to the plane of the structure at the ends, this design will have the same usable space as Design 2. The large radius of curvature in the crossed arch members, however, will allow for very uniform distribution of force through these members.

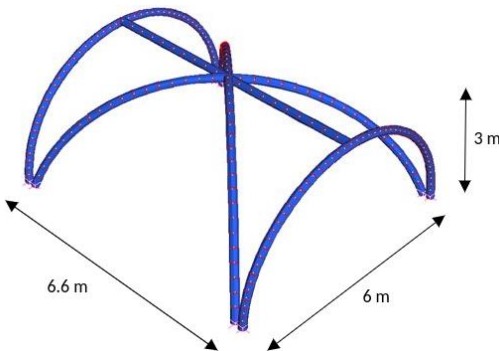

**Figure 4.** Structural system Design 3 isometric view showing dimensions in three orthogonal directions.

**Table 7.** Design 3 Structural System Sketches.

| Plan | Front Elevation | Side Elevation |
| --- | --- | --- |
| 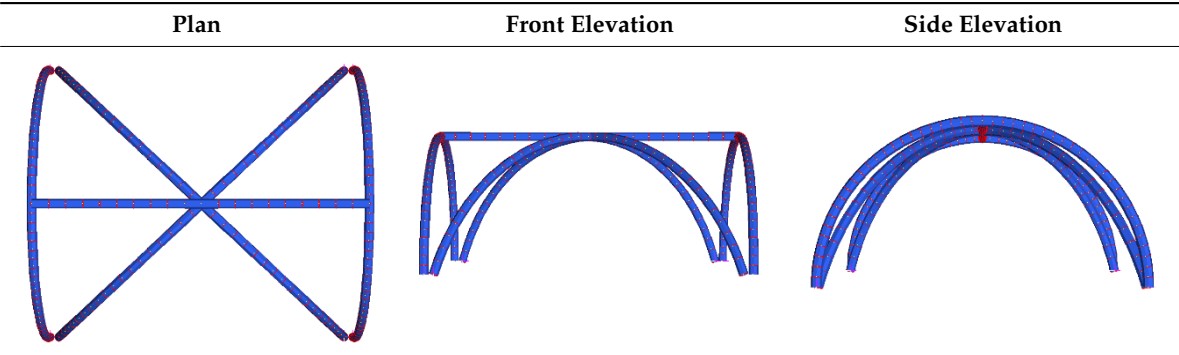 | | |

## 4. The Applied Load

As part of the analysis conducted in this research, the following loads were applied to each design:

- The dead weight of the glass roofing (470 kPa)
- The dead weight of the bamboo sections (425 kg/m$^3$)
- The live load on the floor (0.25 kPa)
- Wind pressures applied based on calculations made for standard conditions in metropolitan Australia, with a 500-year recurrence interval. Due to the curved shape of the structure, the methodology used a combination of AS1170.2 [46] and Eurocode 1 [47].

The pressure provided by the dead weight of the glass roofing is based on 4000 kg/m$^3$ glass, 12 mm thick, which serves as a very conservative value in order to account for varying types of glass used and degree of glazing. This conservative approach allows for the application of this research for a wide range of purposes. Furthermore, the dead weight of bamboo with a density of 425 kg/m$^3$ is also a conservative value, with bamboo typically having a density of 300–400 kg/m$^3$.

The applied live load of 0.25 kPa is typical of roofing, as per AS1170.0 [46], and assumes that there is no access to pedestrians. Within the scope of the analysis in this report, this is a satisfactory approximation. In the case of varying tributary width, the dead and live load forces have been applied to arches in "regions", which take the largest tributary width for that given region as the source of calculation. This is particularly prevalent in the designs that incorporate crossed arches, as the tributary width is maximised at the base, and zero at locations of crossing. This therefore provides a conservative analysis, as the forces applied will exceed the true value. An example of this, from a crossed arch in Design 1, can be seen below in Figure 5.

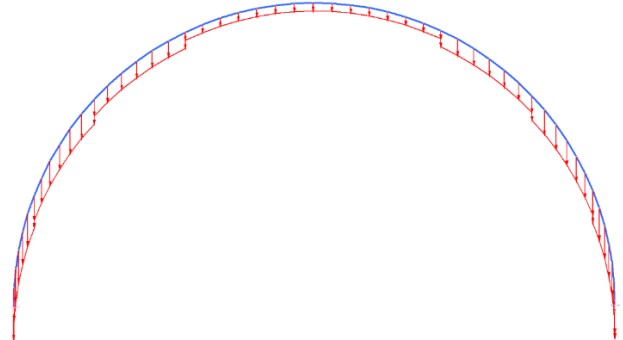

**Figure 5.** Arch loading in Design 1 to account for varying tributary width.

In the case shown in Figure 5, despite the top of the arch being a location of zero tributary width, the design has been constructed to conservatively take the tributary width as the largest value in that region. Also noteworthy are the points in Designs 1 and 3 where there are two arches connected at approximately the same base location. This has a similar loading methodology as shown in Figure 5; however, the smaller tributary width is at the base as opposed to the top. The reason for this conservative approach was due to limitations of the modelling software. When applying distributed loads in Strand7, it is possible to vary the load linearly from node to node. This means to produce a linearly varying load across multiple beam elements, the node to node load change must be manually varied for every single beam element. When modelled with a great number of nodes to obtain an accurate stress distribution, as is the case for all designs in this report, the sheer number of beam elements utilised makes this a very labour and time-consuming process. As a result, this was avoided in favour of a more conservative modelling procedure. To conduct an analysis of each design, combination loading in both serviceability and ultimate limit states from AS1170 [46] were used. The worst-case limiting state was determined for each design and was used as the minimum requirement for section capacity in each design. This allowed for the determination of bamboo construction guidelines within the scope of the Australian Standards. The following five load cases were applied to each design ultimate limiting state including (1.35G; 1.2G + 1.5Q; and 1.2G + W + Q) and serviceability limiting state (G + Q and G + 0.7Q + W).

All designs were constructed using strictly circular arches, resulting in the load path shown in Figure 6. Downward loads applied to the structure will be transferred down through each arch and into the rigid connections at the base, resisting the thrust action of the arch. Alternatively, the load path is reversed for wind loading, due to the suction effect that is magnified by the dome shape. Due to the rigid connections between arches, lateral loading is transferred between arches and then transferred to the base with the same load path as shown.

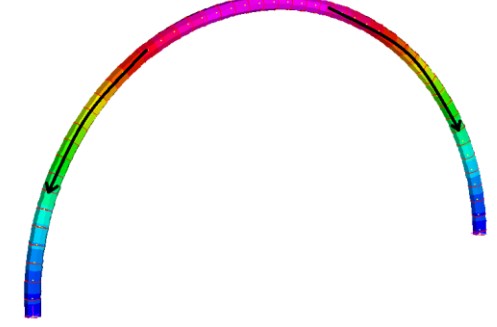

**Figure 6.** Load Path of Singular Modelled Arch for Downward Loading.

## 5. Numerical Analysis

The designs were initially modelled as arched lines in AutoCAD and then imported into Strand7 for numerical analysis as beam elements. This assisted with the modelling of curved elements with each member then subdivided to replicate the configuration of bamboo, connected by nodes at small and regular intervals. This subdivision to replicate the actual material also lowers the potential for numerical error in the solution generated. All members of each design were designed in this manner and were assigned the same material and geometrical properties, as discussed previously in Table 4. Beam elements were selected to model bamboo due to the constraint of pipe elements being unable to be solved in the non-linear solver on Strand7 and the ability to import curved members from AutoCAD. A summary of the material properties of bamboo used in the models can be found in Table 8.

**Table 8.** Material capacities of bamboo (from Table 2).

| Property | Ultimate Strength Capacity (MPa) |
|---|---|
| Tensile Stress | 115.0 (centre), 120.0 (ends) |
| Compressive Stress | 108.0 |
| Fibre Stress | 43.50 |
| Shear Stress | 29.12 |
| Bending Stress | 97.13 |

Prior to the application of boundary conditions, the base nodes had six degrees of freedom, three translational and three rotational. These nodes were fixed in all directions following the application of boundary conditions. This prevented the movement of the base nodes in any direction and the rotation of the structure at the foundations. Fully fixed boundary conditions at the base connection of each member are representative of the realistic foundation conditions which would be in place. The analysis of these designs was conducted using a linear solver to generate displacement, axial stress, bending stress and combined loading. The use of the linear solver was deemed appropriate for these designs, due to the relatively small amount of displacement in each design compared to the overall size. This small displacement means that the non-linear components of stress will be relatively small in comparison to linear components. This is supported by prior research conducted in the analysis of curved structures that indicates that there is, at most, a 10% variation in linear and non-linear values [48]. A flow chart showing the complete methodology of analysis is shown in Figure 7.

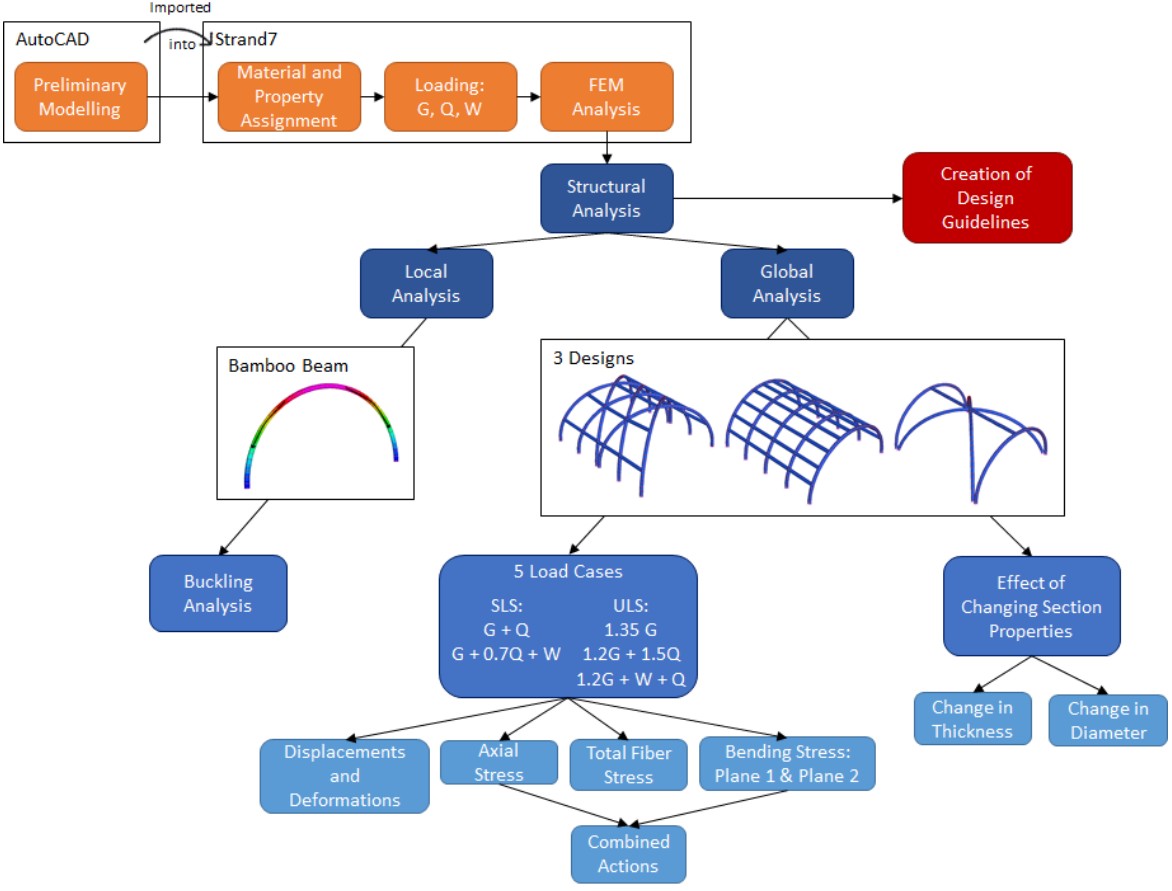

**Figure 7.** The proposed research method.

*Buckling Analysis for Curved Members*

The following section details the buckling calculations for curved bamboo members. In the case of each design, the largest radius arch was chosen for analysis in order to find the most critical buckling load. From Timoshenko's theory of elasticity [49], which states that a curved member with an arched centreline forming part of a circle will buckle as shown by the dashed line in Figure 11, under a critical pressure $q_{cr}$. The following Equation (1), represents the initial circular arch as a function of uniform pressure along the beam, as in the theory.

$$k^2 = 1 + \frac{qR^3}{EI}$$
$$\text{Therefore, } \frac{d^2w}{d\theta^2} + k^2w = 0 \tag{1}$$

The general solution for the equation of buckling of a uniformly compressed circular arch is then provided in Equation (2). These variables are depicted in Figure 11. The variables shown in Figure 8 are used for the calculation of the above Equation (1) and below general form of Equation (2). w is taken to equal the radial displacement towards the centre and θ equalling the degree angle along the beam.

$$w = A \sin k\theta + B \cos k\theta \tag{2}$$

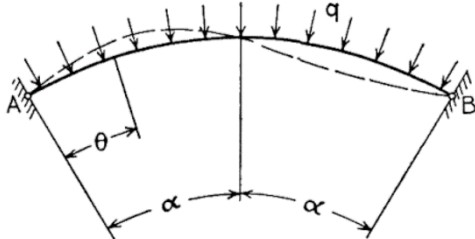

**Figure 8.** Buckling of a uniformly compressed circular arch.

The variables A and B are constants used in Equation (2) and Figure 8 to replicate the boundary conditions at either end of the beam at the points of connection. Other variables depicted include $\theta$, representing the degree angle along the beam; $\alpha$, representing half of the angle of the arched segment; and R, being the radius of the circle equivalent to the arched member. It is important to note that the radius of the arch is not always equal to the height of the arch—this is the case when $\alpha$ is less than $90°$ (or less than $\frac{\pi}{2}$). At the left end of the beam, $\theta$ is taken as 0 and, at the right end, $\theta$ is taken as $2\alpha$. This equation is then taken to satisfy the condition of inextensibility at the centre ($k = \pi/\alpha$). From these values, the critical buckling load can be found, provided in Equation (3).

$$q_{cr} = \frac{EI}{R^3}\left(\frac{\pi^2}{\alpha^2} - 1\right)$$
(3)

The above Equation (3) is representative of the critical pressure ($q_{cr}$) which will induce buckling within the member. The notation used in this equation includes the following variables: $E$ is taken as the elastic modulus of the material; $I$ is taken as the second moment of area for the cross section; R is taken as the radius of the circle equivalent to the arched member; and $\alpha$ represents half of the angle of the arched segment. These calculations assume that the arched member, when bucked, has an inflection point at the middle of the beam, as shown in Figure 8. Although there is a possibility for multiple inflection points symmetrical to the centre of the beam, these types of inflection provide a higher critical buckling load, according to Timoshenko's theory of elasticity. This justifies the use of a single inflection assumption to find the lowest critical buckling load, resulting in a conservative calculation. Another assumption of these calculations is that of a curved beam member before buckling. This assumption requires prestressed members prior to connecting the end supports of the beam, otherwise some initial bending on the members will occur and the beam will no longer be arched uniformly. This condition is assumed to be satisfied through the process of bending the bamboo members into their curved form. In the case of the selected designs, the values to be used in the calculation above are given in Table 9, which is calculated from the properties and dimensions of the designs.

**Table 9.** Values Required for Critical Buckling Load per Curved Member.

| Values | Design 1 | Design 2 | Design 3 |
|---|---|---|---|
| E | $1.5 \times 10^{10}$ Pa | $1.5 \times 10^{10}$ Pa | $1.5 \times 10^{10}$ Pa |
| I | $3.14 \times 10^{-5}$ m$^4$ | $3.14 \times 10^{-5}$ m$^4$ | $3.14 \times 10^{-5}$ m$^4$ |
| R | 3.43 m | 3.00 m | 4.71 m |
| $\alpha$ | $\frac{\pi}{2}$ | $\frac{\pi}{2}$ | $\frac{\pi}{2.6202}$ |
| **Critical Buckling Load** ($\mathbf{q_{cr}}$) | 35.02 **kN/m** | 52.33 **kN/m** | 26.44 **kN/m** |

Table 9 shows the critical buckling load for a uniformly distributed load on each of the designs. The main varying value for each design is the value of the largest member curvature R. As shown in Equation (3), R is cubic inversely proportional to the buckling load. Consequently, the smaller the radius of curvature for a member, the higher the critical buckling load. Design 2 has the smallest radius of curvature for its members and, therefore, the highest critical buckling load. Similarly, Design 3 has the largest radius of curvature for its members and, therefore, the lowest critical buckling

load. Although constant between each design, a larger second moment of area I will increase the critical buckling load for a section. Due to the relatively large second moment of area of the hollow circular cross section of bamboo, the use of this material assists with resisting buckling. As these calculations have been made using Timoshenko's theory of elastic stability, they overestimate the member's ability to resist buckling. This is due to the approximation of elasticity not accounting for the uneven distribution of stresses that occurs in plastic deformation. On the other hand, there has been no consideration of restraint due to crossed-arches, lateral supports and connection points for the glass, nor additional capacity from bamboo diaphragms. All these additional sources of restraint will reduce the effective length for buckling and contribute to a greater critical buckling load for the bamboo members. Despite the approximations used in these calculations, the critical buckling load of each design greatly exceeds the loads applied and, thus, it is a reasonable assumption that these sections will not undergo out-of-plane buckling.

## 6. Results

For each design, two serviceability and three ultimate limiting state load cases determined from AS1170.0 [46] are presented below. The critical case for both serviceability and ultimate strength is deemed based on the maximum value of displacement and for each individual stress, which are considered independently of each other. For all cases, the results shown are from designs with 70 mm diameter and 12 mm thickness bamboo sections.

### 6.1. Displacement

The displacement of a design is the measure of movement from the original position of the structural members. The deformation results were achieved by the numerical analysis in Strand7 as detailed displacement results under different load cases. A summary of these results is provided in Figure 9.

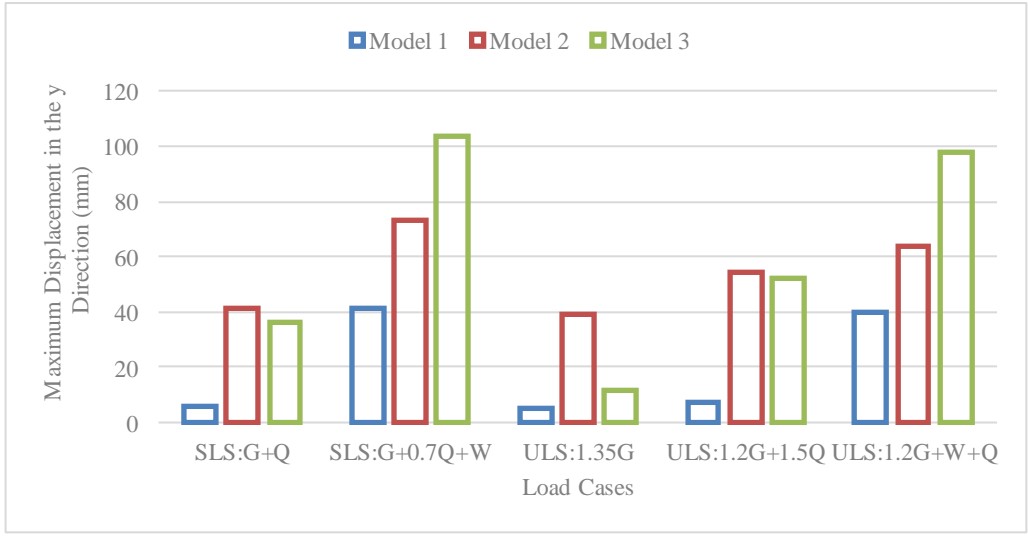

**Figure 9.** Summary of maximum displacement results for differing load cases and designs.

In Figure 9, wind is shown to be the critical load in both serviceability and ultimate limit states, with the greatest deflection in load cases featuring wind loading. Design 1 can be seen to have the smallest maximum displacement for all load cases, indicating that it provides the best distribution of forces to minimise deformation. Notably, Design 2 is seemingly susceptible to deformation under dead and live loads, being the design with the largest deformation in load cases that omit wind loading. Finally, the limited lateral restraint from cross-arches or horizontal bamboo members leads to great deformation in Design 3 from wind loading. Table 10 shows the physical displacement of the structures

under the greatest load case (SLS: G+0.7Q+W) with a displacement of 5%. Table 10 shows that for each design the top right face is the location of maximum deformation. This is the windward face of each model, where the wind pressure is both largest and in the direction of dead and live loads. As anticipated, this is particularly pronounced in Design 3, which has the least members and therefore largest tributary area.

**Table 10.** Displacement of designs under SLS: G+0.7Q+W (5% deflection).

| SLS: G+Q | Model 1 | Model 2 | Model 3 |
|---|---|---|---|
| Y<br><br>X | Max Displacement = −5.89 mm | Max Displacement = −41.64 mm | Max Displacement = −36.36 mm |
| **SLS: G+0.7Q+W** | **Model 1** | **Model 2** | **Model 3** |
| | Max Displacement = 41.28 mm | Max Displacement = 73.28 mm | Max Displacement = 103.7 mm |

## 6.2. Axial Stress

The measure of axial force in each of the members creates internal forces within the beam measured as axial stress. These stresses can be either in tension or compression and both types of stresses can be

presented in the detailed results. Figure 10 displays a graphical summary of the greatest axial tension stress throughout the structure and Figure 11 portrays the greatest axial compression stress.

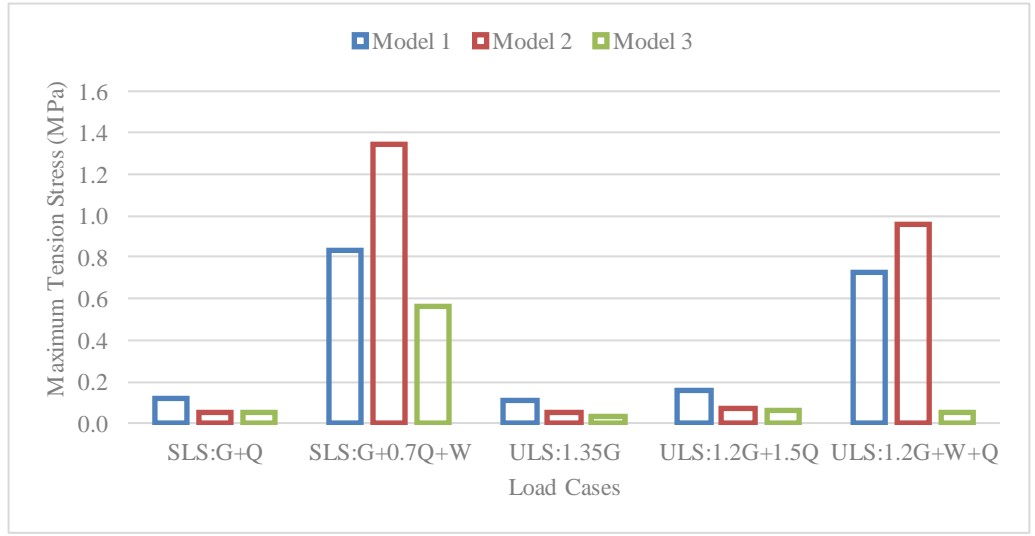

**Figure 10.** Summary of maximum tension stress for differing load cases and designs.

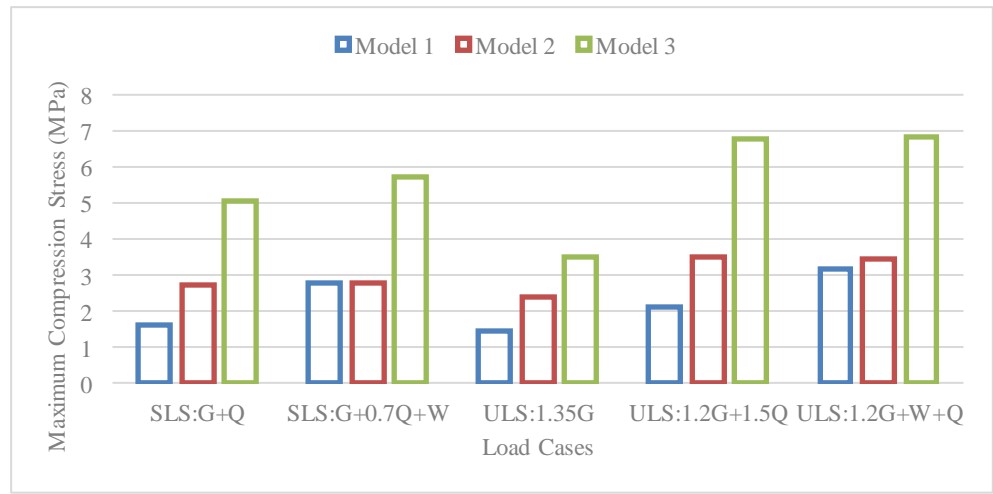

**Figure 11.** Summary of maximum compression stress for differing load cases and designs.

The tension stresses in each design are mostly due to wind loading, as shown in Figure 10, with only some occurring in the dead and live load cases. This is due to dead and live load being a purely downward load, causing only minor tension stresses at base restraint locations. It is notable that for these downward cases, the tension stresses are largest for Design 1, which is a result of the reliance on cross-arches, not horizontal members, to distributed forces throughout the structure. The reliance on cross-arches therefore generates more locations of large thrust, which require restraint at the base. Alternatively, wind loading is a lateral load and therefore causes tension stresses in the leeward face of each design. Consequently, tension stresses are maximised in load causes that have wind added. Despite this, the relative magnitude of these tension stresses is low, as the leeward wind pressures must oppose dead and live loads, and in no case does the maximum tensile stress exceed bamboo's capacity of 115 MPa. Figure 11 shows that most compressive stresses in the designs comes from dead and live loading, with only a relatively small increase in stress with wind loading. Notably, under wind loading, Designs 1 and 2 have comparable maximum compressive stresses on the windward face, despite Design 1 having lower maximum compressive stress for dead and live loads only. This is due

to the larger tributary width of members in Design 1 in wind design, arising from the orientation of crossed arches, making it more susceptible to wind loading. Additionally, Design 3 has relatively much higher maximum compressive stress than the other designs, but in all cases this stress does not exceed bamboo's compressive capacity of 108 MPa.

When comparing Figure 10, Figure 11 and Table 11, the relative maximum stresses are larger for compression than tension. This is due to the tensile stresses occurring only due to leeward wind pressures having to oppose the purely compressive loading of dead and live loads, or because of fixed base restraint. Alternatively, the maximum compressive loading occurs when windward pressures load in the same direction as dead and live loads and is therefore relatively larger. Therefore, when considering axial stresses for serviceability and ultimate loading, a consideration of compressive stress is much more important than its tensile counterpart.

**Table 11.** Beam axial stress results from Strand7 under different load conditions.

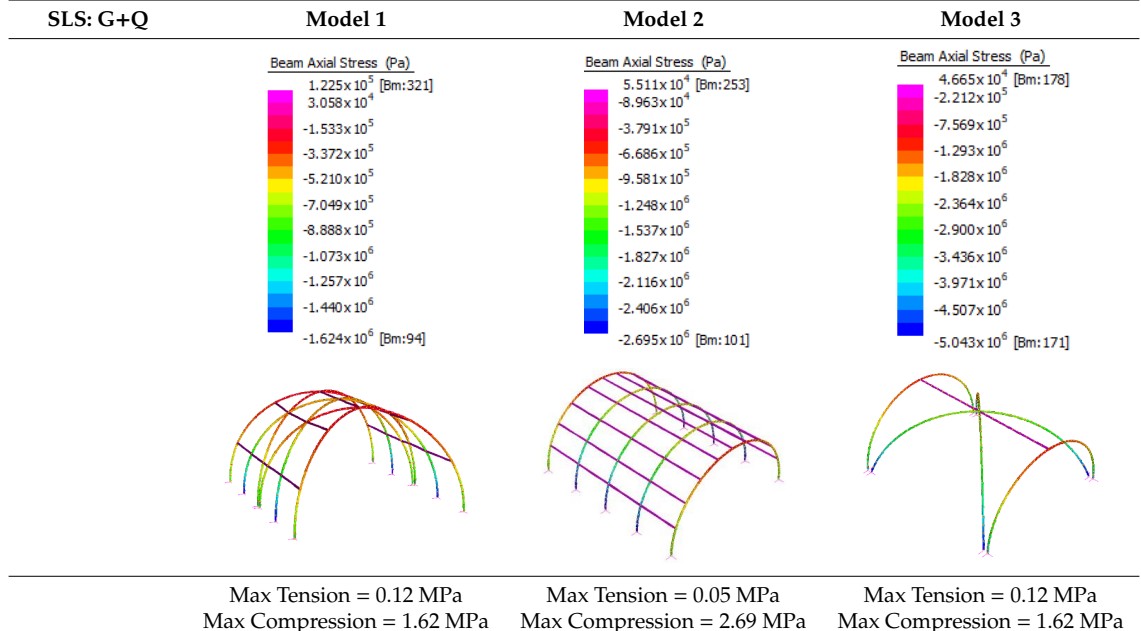

| SLS: G+Q | Model 1 | Model 2 | Model 3 |
|---|---|---|---|
| | Max Tension = 0.12 MPa | Max Tension = 0.05 MPa | Max Tension = 0.12 MPa |
| | Max Compression = 1.62 MPa | Max Compression = 2.69 MPa | Max Compression = 1.62 MPa |

### 6.3. Total Fiber Stress

The total fibre stress is representative of the stresses on the cross section of the beam from the combination of bending moments in both planes and the axial force. A consequence of fibre stress is that it is dependent on location in the cross-section—that is, the fibre stress for a given cross-section will be different at the top and bottom extremities. Figure 12 indicates that there is exceedance of the 43.5 MPa limiting capacity of bamboo in one serviceability and two ultimate limit states for Designs 2 and 3 for fiber stress. The resulting failure in these specific fibers will result in uneven stress distributions, and likely further failures to catastrophic consequence for structural stability. This, however, is not reflected in the models, as they are not specifically limited by the capacity of bamboo. As a result, it cannot be confirmed whether Designs 2 and 3 are stable under fiber stress considerations, and only Design 1 provides an assuredly suitable distribution of fiber stress to avoid failure. It is notable that the maximum magnitude of fiber stress shown in Figure 12 and Table 12 came from compressive fiber stress in each case. The curvature of the bamboo members results in a tensile fiber stress at the top and compressive at the bottom of equal magnitude, which is opposed in magnitude by the loading mechanisms. Therefore, the maximum tensile fiber stress is at most equal to the compressive fiber stress without any loading; however, with mostly compressive loading, as in this analysis, the compressive fiber stress will exceed this value.

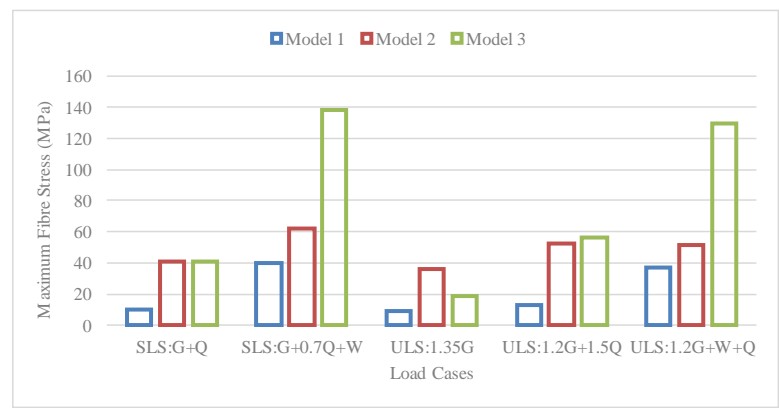

**Figure 12.** Summary of maximum magnitude fibre stress results in compression for differing load cases and designs.

**Table 12.** Total beam fibre stress results.

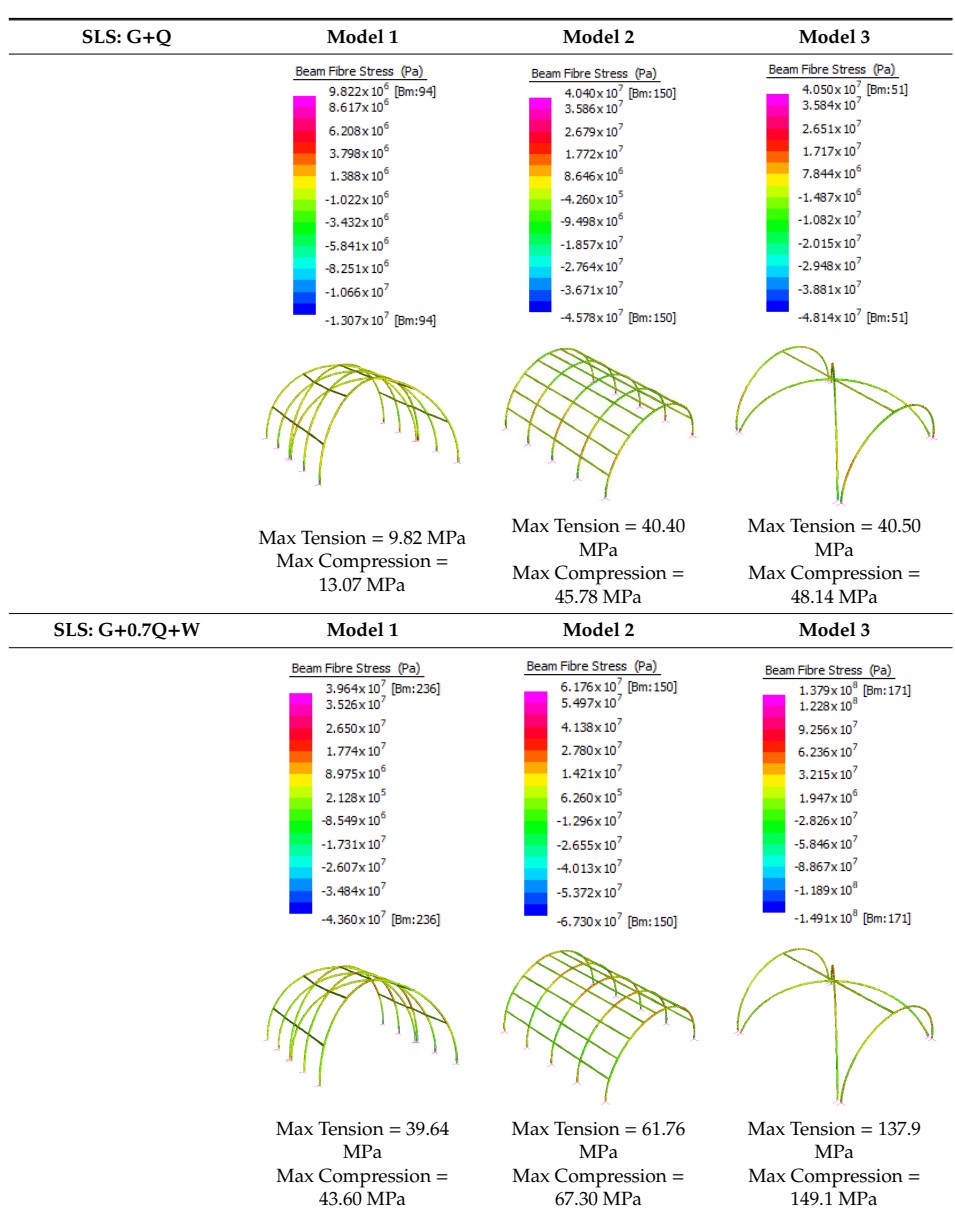

| SLS: G+Q | Model 1 | Model 2 | Model 3 |
|---|---|---|---|
| | Max Tension = 9.82 MPa Max Compression = 13.07 MPa | Max Tension = 40.40 MPa Max Compression = 45.78 MPa | Max Tension = 40.50 MPa Max Compression = 48.14 MPa |
| **SLS: G+0.7Q+W** | **Model 1** | **Model 2** | **Model 3** |
| | Max Tension = 39.64 MPa Max Compression = 43.60 MPa | Max Tension = 61.76 MPa Max Compression = 67.30 MPa | Max Tension = 137.9 MPa Max Compression = 149.1 MPa |

*6.4. Bending Stress*

Bending stress analysis was conducted along two planes, about the plane of the beam element (plane 1) and about the plane perpendicular to the beam element. The maximum magnitude bending stresses about plane 1 are summarised in Figure 13 for each design and load case. Figure 13 indicates that for bending stresses the bamboo in Design 3 will fail under some serviceability and ultimate limiting states. This is due to the bending capacity of bamboo, 97.1 MPa, being exceeded. As a result, Design 3 will have catastrophic failure of the member, and consequently the structure itself.

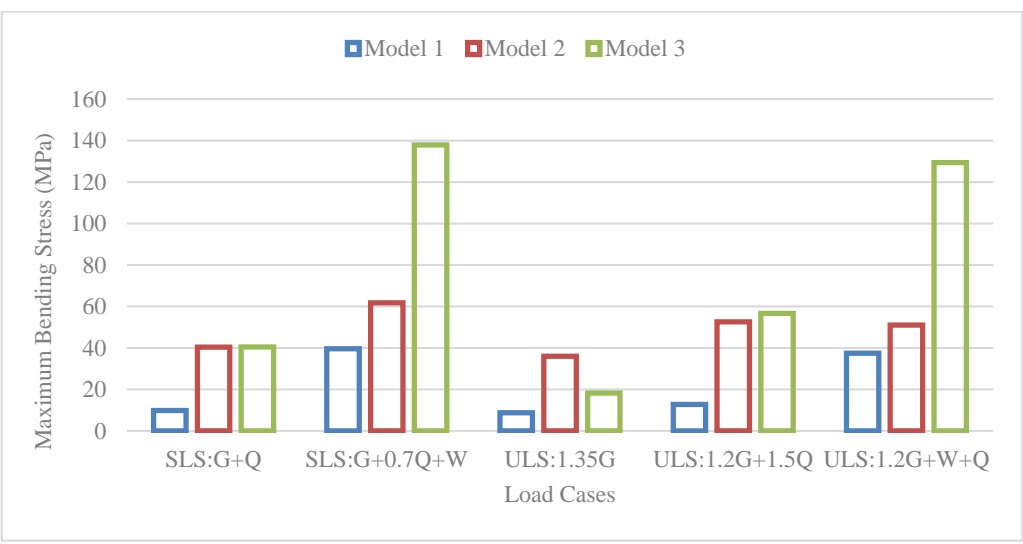

**Figure 13.** Summary of maximum bending stress results for differing load cases and designs.

Alternatively, Design 1 and 2 sufficiently distributed bending stresses throughout the structure as to avoid failure of the bamboo. In all cases, the cross-arches used in Design 1 result in much lower bending stresses compared to other designs. The relative magnitude of this lower result is minimised in load cases with wind loading when compared to Design 2. This is reflected by the relatively larger increase in bending stress with the addition of wind loading for Design 1 than for Design 2. This is to be expected, due to the larger tributary width in the specific design of Design 1 having a larger susceptibility to wind pressure and thus remains the most effective design for the minimisation of bending stress.

*6.5. Combined Action*

The combined action loading consists of the combination of axial stress, bending in plane 1 and bending in plane 2. This allows analysis of the structures in a more realistic loading situation, where all stresses will be simultaneously acting. Figure 14 and Table 13 details the maximum stress under combined action below. Figure 14 indicates that in all load cases, Design 1 distributes force through the structure best, seemingly uniformly distributing bending and axial stresses effectively. Alternatively, Designs 2 and 3 show a lower capacity to distribute stress, with Design 2 being particularly susceptible to dead and wind loading, and Design 3 susceptible to live and wind loading. In comparison to Figure 15, where Designs 2 and 3 show a similar response to bending stress, Figure 14 indicates that Design 3 in most cases has a higher combined action stress. This is due to the compressive stresses in this structure, indicated by Figure 11 to be much larger than for Design 2. Seemingly, this means that Design 3 is the least stable of the designs.

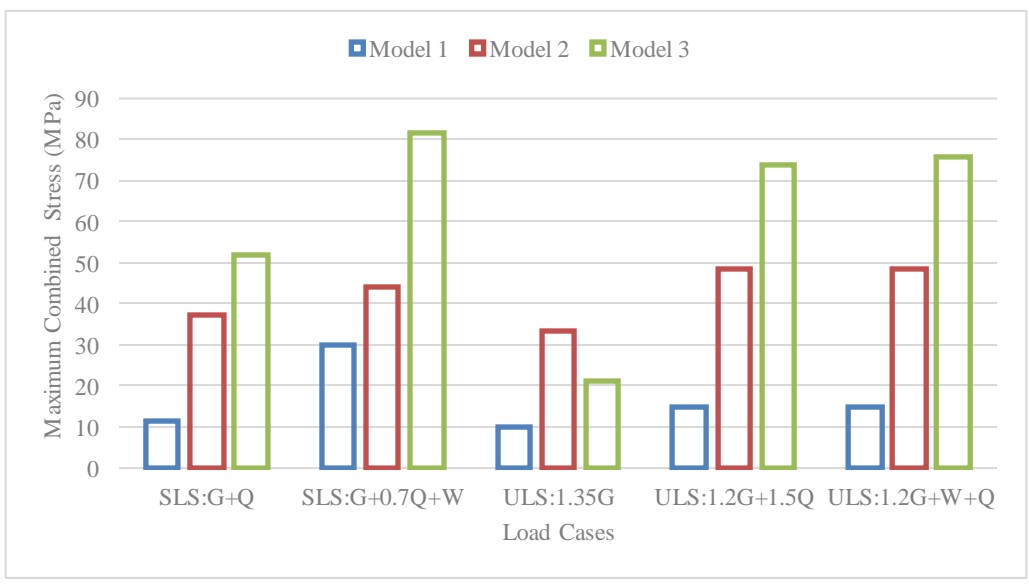

**Figure 14.** Summary of maximum combined action loading results for differing load cases and designs.

**Table 13.** Total combined actions (bending in both directions and axial stress).

| SLS: G+Q | Model 1 | | Model 2 | | Model 3 | |
|---|---|---|---|---|---|---|
| | | MIN | MAX | | MIN | MAX | | MIN | MAX |

| SLS: G+Q | | Model 1 | | | Model 2 | | | Model 3 | |
|---|---|---|---|---|---|---|---|---|---|
| | | MIN | MAX | | MIN | MAX | | MIN | MAX |
| | BM1(N.m) | $-3.057 \times 10^2$ [Bm:94] | $2.853 \times 10^2$ [Bm:47] | BM1(N.m) | $-1.180 \times 10^3$ [Bm:101] | $1.180 \times 10^3$ [Bm:150] | BM1(N.m) | $-1.214 \times 10^3$ [Bm:51] | $1.214 \times 10^3$ [Bm:50] |
| | BM2(N.m) | $-5.920 \times 10^1$ [Bm:95] | $8.412 \times 10^1$ [Bm:46] | BM2(N.m) | $-5.645 \times 10^1$ [Bm:268] | $5.383 \times 10^1$ [Bm:80] | BM2(N.m) | $-2.085 \times 10^2$ [Bm:165] | $4.948 \times 10^2$ [Bm:171] |
| | AxForce(N) | $-3.552 \times 10^3$ [Bm:94] | $2.679 \times 10^2$ [Bm:321] | AxForce(N) | $-5.893 \times 10^3$ [Bm:101] | $1.205 \times 10^2$ [Bm:253] | AxForce(N) | $-1.103 \times 10^4$ [Bm:171] | $1.020 \times 10^2$ [Bm:178] |

**Model 1**　　　　　　　　**Model 2**　　　　　　　　**Model 3**

| SLS: G+0.7Q+W | | Model 1 | | | Model 2 | | | Model 3 | |
|---|---|---|---|---|---|---|---|---|---|
| | | MIN | MAX | | MIN | MAX | | MIN | MAX |
| | BM1(N.m) | $-1.140 \times 10^3$ [Bm:236] | $6.670 \times 10^2$ [Bm:227] | BM1(N.m) | $-1.768 \times 10^3$ [Bm:150] | $1.279 \times 10^3$ [Bm:139] | BM1(N.m) | $-2.725 \times 10^3$ [Bm:171] | $1.092 \times 10^3$ [Bm:40] |
| | BM2(N.m) | $-2.781 \times 10^2$ [Bm:70] | $2.921 \times 10^2$ [Bm:118] | BM2(N.m) | $-1.337 \times 10^2$ [Bm:80] | $1.337 \times 10^2$ [Bm:180] | BM2(N.m) | $-2.844 \times 10^3$ [Bm:171] | $1.582 \times 10^3$ [Bm:129] |
| | AxForce(N) | $-6.123 \times 10^3$ [Bm:46] | $1.814 \times 10^3$ [Bm:106] | AxForce(N) | $-6.120 \times 10^3$ [Bm:100] | $2.941 \times 10^3$ [Bm:138] | AxForce(N) | $-1.246 \times 10^4$ [Bm:100] | $1.228 \times 10^3$ [Bm:39] |

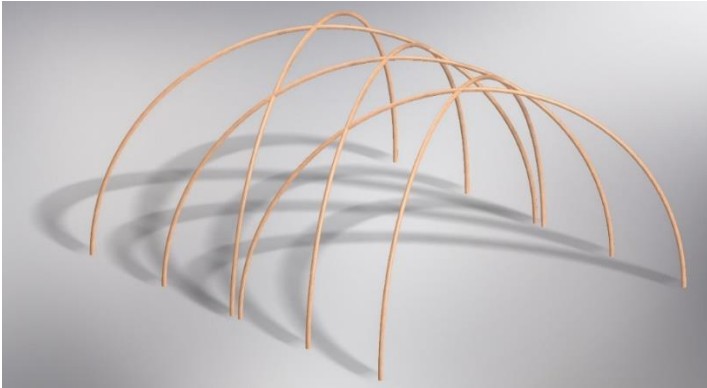

**Figure 15.** Suitable bamboo structural design solution (rendered design).

Without a defined failure stress for combined action, there is no definitive way to determine the structural stability of any design. Figure 14 does, however, explicate that Design 1 is most effective in distributing stresses under typical AS1170 [46] loading conditions. From this, it is expected that it is the most stable of the designs, capitalising best on the design characteristics of curved arches, and bamboo's inherent properties.

## 7. The Effect of the Section Properties on the Structural Responses

To investigate the impact of varying properties of bamboo sections, the maximal stress and displacement values in Design 1 were recorded for a range of section designs. These sections included three different thickness with the same diameter (D = 70 mm) and three different diameters with the same thickness (t = 12 mm), as per Table 14.

**Table 14.** Section Classifications and Changing Properties.

| Classifications | Dimension | Load Combination |
|---|---|---|
| Section 1 | • Diameter: 70 mm<br>• Thickness: 8, 12, 18 m | |
| Section 2 | • Diameter: 90, 130, 170 mm<br>• Thickness: 12 mm | • Ultimate Limit State:1.2G+W+Q |

The below Table 15 shows the results from analysis conducted on changing material properties. This analysis was performed using the initial models and method demonstrated in the Models and Numerical Analysis sections previously in this report. The initial size, which was assumed for the model comparison previously, is varied in both bamboo diameter and thickness, to analyse the associated effects on the stresses and displacement of members in the structures. The maximal stresses and displacements in Design 1 for these changing section dimensions, under load combination ULS 1.2G+W+Q, can be seen in Table 15. The relative changes in the properties compared to the values directly above are shown in parentheses.

**Table 15.** Summary of analysis results for changing beam sections Design 1 (ULS 1.2G+W+Q loading only).

- **Section 1—Diameter 70 mm; Thickness 8 mm, 12 mm, 18 mm**

| Thickness (mm) | Stress (MPa) | | | Bending Stress (MPa) | | Displacement (mm) |
|---|---|---|---|---|---|---|
| | Axial | | Fibre | Bending Plane 1 | Bending Plane 1 | |
| | Tension | Compression | | | | |
| 8 | 1.026 | 4.428 | 53.20 | 50.13 | 13.30 | 50.86 |
| 12 | 0.725 | 3.169 | 41.83 | 39.64 | 10.48 | 40.19 |
| 18 | 0.536 | 2.369 | 35.61 | 33.97 | 8.96 | 34.43 |

- **Section 2—Diameter 90 mm, 130 mm, 170 mm; Thickness 12 mm**

| Diameter (mm) | Stress (MPa) | | | Bending Stress (MPa) | | Displacement (mm) |
|---|---|---|---|---|---|---|
| | Axial | | Fibre | Bending Plane 1 | Bending Plane 1 | |
| | Tension | Compression | | | | |
| 90 | 0.526 | 2.360 | 19.51 | 21.16 | 5.604 | 16.79 |
| 130 | 0.329 | 1.563 | 7.653 | 8.773 | 2.333 | 4.893 |
| 170 | 0.236 | 1.167 | 3.846 | 4.703 | 1.257 | 2.042 |

Section 1 of Table 15 shows the corresponding results of changing the thickness of a 70 mm diameter section for Design 1. The values were increased from 8 mm to 12 mm and then further to 18 mm, analysing and logging the results of the output on the model. Section 2 details similar results, but with a constant thickness of 12 mm and changing diameter properties from 90 mm to 130 mm, then to 170 mm. These results were taken for axial stress, fibre stress, bending stress and displacement.

Based on Section 1 of Table 15, increasing the thickness has a greater effect on the section's capacity to control axial stresses, with a relatively lower reduction of fibre stress, bending stress and displacement. This is a consequence of the increase in section area due to increasing thickness, albeit a relatively marginal increase in second moment of area. A further consequence of this is that the relative change in stress and displacement decreases for larger thicknesses.

This means that a 50% increase in thickness for a thin section will have a larger relative impact on section capacity than a 50% increase for a thick section. Therefore, a larger thickness section is desirable in primarily axially loaded sections; however, the design must be wary of the limited degree of stress reduction for only changing thickness.

The dependence of fibre stress, bending stress and displacement on the second moment of area is shown by increasing the diameter of the section in Section 2 of Table 15. An increase in diameter of the section by 44% leads to a decrease in fibre stress, bending stress and displacement of 61%, 58% and 71%, respectively. There is also an increase in the section's area due to a larger diameter, providing a similar axial stress reduction as thickness increases. Also noteworthy is that the percentage decrease in stress and displacement is proportional to the initial diameter. For example, the 44% increase in diameter from 90 mm to 130 mm corresponds to a 1.34% decrease in bending stress per 1% diameter increase, whilst the increase in diameter from 130 mm to 170 mm corresponds to a 1.48% decrease in bending stress. This means that the relative decrease in stress and displacement with respect to increasing diameter is larger for larger diameter sections. The difficulty of obtaining larger diameter bamboo sections, however, indicates that these sections are more favourable in situations of larger fibre stress, flexure and displacement–and thicker sections for axially loaded members.

Despite the results shown in Table 15 corresponding to changing section dimension, this is more representative of changing the number of bamboo sections utilised in a member. The increased area

and second moment of area from binding together multiple bamboo sections will have an equivalent reduction in stress and displacement as that seen for increased dimensions. This is a more realistic and achievable means to control these properties in design.

## 8. The Preliminary Design Guidelines for Bamboo

As a result of the analysis in this study, when considering designs using bamboo members, there are elements that must be considered. With respect to section dimension, larger thickness bamboo is desirable for primarily axially loaded situations, whilst larger diameter bamboo is desirable for control of displacement, fibre stress and bending stress. Therefore, straight bamboo members are practical in the context of columns and horizontal lateral support systems.

Alternatively, there is better transfer of force between bamboo members when utilising curved members. The use of crossed arched structures in bamboo provides greater transfer of lateral loads and more uniform stress distribution than conventional straight members. The consideration, however, is that the uniformity of stress distribution is maximised by increasing the radius of curvature, whilst simultaneously increasing the susceptibility to out-of-plane buckling. Moreover, increasing the radius of curvature for an arch will increase the usable space for a structure. The design of a curved bamboo arch shape must then find a radius of curvature which maximises stress distribution uniformity and usable space without risking buckling.

With this information, it is possible to manipulate the desired area and second moment of area of a bamboo member through binding together multiple bamboo sections. This provides a more practical means to increase capacity than relying on bamboo growing with larger diameter or thickness and is an important consideration for design with bamboo.

Despite the care taken to ensure the validity of investigation in this report, there are elements in the modelling that would benefit from improved experimental design. The use of Strand7 as the finite element modelling program produced two main limitations, specifically the approximation of the member curvature and the inability to include failure stress limits. The curved bamboo members in each model were constructed from multiple finite straight beam elements, as opposed to a true curved shape. Particular attention was given to increasing the frequency of nodes along curved members to minimise this approximation; however, ultimately the software was unable to provide a true representation of reality. There exists finite element modelling software that allows for the construction of curved members through either rotated extrusions of a surface, or construction of elements by sweeping. These model mechanisms would have produced a shape much more consistent with reality, therefore becoming a much more accurate source of analysis. Despite this, the use of such programs as Abaqus proved disadvantageous for the scope of this report, due to the difficulty of design models requiring multiple planes and the inability to import the models from AutoCAD. Furthermore, Strand7 prevents the input of failure stresses as a limiting property of an element, which is otherwise available in other software. By imposing failure stresses onto a material in finite element modelling, a solver can determine when a member undergoes catastrophic failure. This provides a much more accurate understanding of failure mechanisms when analysing a design. For example, in this report, where Designs 2 and 3 exceed fibre stress and bending stress limits under certain load cases, Strand7 does not unload these members, assuming a continued elastic loading. Therefore, the stress these members are subjected to is beyond their capacity and consequently a false representation of the structure's stability. In reality, exceedance of this stress would cause localised member failure and require the stresses to be distributed elsewhere in the structure.

The global modelling used in this report utilised circular hollow beam sections to represent bamboo, which did not allow for any inclusion of diaphragm components. Due to the grain of bamboo diaphragms being perpendicular to the grain of the member, this was beyond the capabilities of Strand7. To accurately model this component of bamboo, a more sophisticated software must be used. Despite this limitation, the diaphragm is projected based off research conducted within the literature review, to increase the capacity of the bamboo member compared to a circular hollow section. This means that

the modelling used in this report is representative of a more conservative analysis, and there is no loss of validity in the research. The benefit therefore of including diaphragms would not be to increase experimental validity, but to have a more exact quantification of bamboo's capability in construction.

There is a limitation on the number of designs used in this report, purely as an attempt to focus the scope of analysis. The result of fewer designs is that conclusions are made based only a few dissimilar global aspects of each design, such as the lateral restraint being predominately cross-arches or horizontal members. When preparing more specific design guidelines, however, there is much more to be gained from more subtle changes in designs, such as the number of restraints or the angle between arches at intersections. With these subtle changes, there is potential for much greater analysis; however, this is with the addition of considerable modelling and breadth of scope. To produce a preliminary set of design guidelines, however, the modelling in this report is satisfactory.

Ultimately, this report lacks any validation of models through experimental investigation. As bamboo is a relatively unexplored material, analysis purely by modelling assumes that bamboo behaves as predictably as other typical construction materials such as timber. The diaphragm component in itself is highly untypical, however, and without experimental validation of at least one model used in this report, there is no confirmation of true validity. To control this, the methodology intentionally maintained a conservative approach where possible, aiming to account for any undiscovered discrepancies between bamboo and timber behaviour.

## 9. Conclusions

This paper detailed potential design solutions used to implement a bamboo grid shell structure. The paper aimed to identify how bamboo can be used in building construction. Three designs were proposed and then analysed using finite element modelling to estimate bamboo's strength and serviceability limits and to determine favourable design characteristics for bamboo. The designs utilised similar grid-like structures formed of curved bamboo members combined with thin curved glass used as cladding. Each design varied, however, in member positioning and lateral bracing, to allow a comparison between them.

The most functional design solution based on the results presented throughout this report was found to be Design 1. This design utilised the cross-over of bamboo members resulting in longer members but greater lateral stability than the alternative models. As shown in Figure 15, the use of crossed-arch members allowed for the better distribution of stress and reduced displacement. This design aspect allowed for greater performance in all categories analysed, including deflection, axial stress, bending and combined loading in multiple different load cases compared to the other proposed designs. As a result, Design 1 featured relatively less bamboo compared to the other designs with respect to its mechanical performance. By comparison, the other designs utilised typical horizontal lateral bracing or fewer points of crossed-arch restraint, to the detriment of overall structural performance. This comparison found that larger radius arches, as is a consequence of cross-arch designs, reduces the critical buckling load but greatly enhances overall stability. Therefore, when considering designs with curved bamboo members, provided members will not buckle, crossed-arches serve as a great means of uniform stress distribution. Moreover, the novel analysis conducted in this report finds that bamboo has great potential for future innovation in design, due to its high strength to weight ratio, unconventional timber cross-section and atypical grain orientation.

The effect of changing the material properties is also discussed and it is found that the increase in section size greatly reduces the maximum deflection, bending and fibre stress of the structure, specifically for increased bamboo diameter. Reduction in the axial compression and tension stresses is also noted when increasing the section thickness and diameter. It can then be concluded that members of greater area and I value will increase the loading capacity of the structure, with an increased thickness being suitable for axial loading situations and increased diameter for flexural situations. An alternative method of increasing the area and I value of the bamboo members is the combination of more than one bamboo stalk tied together to form a larger beam. The potential for this option provides an opportunity

for further research and development of completely fixed section to section connections. Despite this, these preliminary design guidelines serve as impetus for future development of more comprehensive bamboo design information.

Some improvements for further research into this area of study have also been identified, with the main focus being on an alternative modelling analysis program. Due to the constraints on this project, Strand7 was the most feasible software to conduct the analysis which served to provide an initial insight into the potential of bamboo as a structural material. Based on these findings for further detailed research, a program which is able to model curved members with diaphragm components and the ability to impose failure stresses is recommended.

**Author Contributions:** Conceptualisation: All authors; Literature review: All authors; Research Method: All authors; Software: R.M., E.A., G.H.J., M.P.J., F.A.M and F.T. Drafting the article based on the numerical reports: F.T. and S.S.; Resources and interpretation: F.A.M. and F.T. All authors have read and agreed to the published version of the manuscript.

**Funding:** This research received no external funding.

**Conflicts of Interest:** The authors declare no conflict of interest.

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
