# Peer review of "Development of Preliminary Curved Bamboo Member Design Guidelines through Finite Element Analysis"

_sustainability, doi:10.3390/su12030822_

Round 1

Reviewer 1 Report

The authors has described a brief guide and conclusions about designing bamboo structures using Finite Element Analisys. The literature supports the methods conclusions. Nevertheless, the structure of the paper and the conclusions could be improved.

Some constructive comments and changes have been proposed to improve some specific points.

General comments:

Try to reformulate the abstract, making it more concise and clear, describing the points sequentally. See the first specific comments. Try to focus the paper on the method to design bamboo structures.The structure of the paper must be improved. The main objective is to develop a guidelines to design bamboo structures. Try to focus the paper in this direction. After finish the "Background review", the methodology should be described in detail. Other points such as "3. Numerical Modelling Strategy", "4. The Applied Load" and "5. Numerical Analysis" should be described into the Methodology. The numerical modelling strategy and the numerical analysis could be unified to describe the CAD models (Designs 1 to 3) and the Finite Elements Analysis details. The point "7. The Preliminary Desgin Guidelines for bamboo" could be included in the conclusion point, once the guidelines is properly described in the methodology. The Figure 7 is your proposed method. It should be included and described as one of the most highligted content of your Metodology point. It is recommended to use passive perfect tenses instead present. See line 351 as an example ("has been constructed" instead "are constructed").

Specific comments:

Line 20. Do not use capital letter in "Bamboo". Line 24. In the abstract, the sentence "To address the issue, ... support system" should be moved after the sentence "These guidelines ... flexural situations". The main objective of this paper is to develop the design guidelines, while the finite element analylsis is a tool to aid the design. Line 26. The redundant sentence "Bamboo is known as ... than timber" could be combined with the first sentences in the abstract. Line 59. Remove "all three preliminary". Your method should be applicable to "any" design, not only for the three selected preliminary designs. Line 60. Sofware AutoCad and Strand7 should include a reference. Line 64. Do not use capital letter in "Bamboo". Line 69. Remove the contraction in "bamboo's use". Line 71. Remove the extra point after the reference. Line 89. Rewrite the inconsisten sentence "Bamboo selected as the main ..." Table 1. Verify and contrast the range of Poisson's Ratio (up to 0.52 > 0.5) from the reference. Use another reference if necessary. Table 1. Try to unify the length units (mm, cm) Line 105. Move "From" to the next line. Line 138. Move "T" to the next line. Line 149. Remove the apostrophe. Line 175. Put "however" at the beginning of the sentence or rewrite the sentence, as more formal style. Line 184. At the end of the sentence "and bolted connections, with two options." What are the two options? If it refers to binding and bolted, try to reformulate the sentence.
Line 193. Missing reference of Eurocode. (Line 193, Line 194, ...). Figure 1. Include the diameter dimension as drawing standars. Include the diamenter symbol and the diametral arrow. Line 223. Use "input" instead "inputted". Line 224. References could be merged as [25-27]. Line 225. Rewrite "These values ... into Strand7" as "These values were applied to the beam elements in Strand7" Table 3. Use GPa as unit for Elastic Modulus (E) as Table 1. Table 3. Use the same Diameter (D) units as Table 1. Table 3. Use the same Thickness (t) units as Table 1. Line 310. The dead weight of the glass is a "weight". Why it is applied as pressure? Pressure acts in a normal direction, while weight "follows" the gravity direction. Line 311. The dead weight of the bamboo is described as a density. Use daN instead kg. Line 315. Standars AS1170.1 and Eurocode 1 should be referenced. Line 323. Standars AS1170.0 should be referenced. Line 333. Figure 8 appears before Figure 7 and Figure 6 in the text. Check if it refers to the Figure 7 in page 13. Move this figure frontwards if so. Figures should appear in the text in a sequential order. Line 349 - Line 350. Live, dead and wind loads combinations have been abbreviated as G, Q, W. It could not be clear for a non-expert reader. Include the meaning of these abbreviations. Line 363. What is the element size (i.e. length) for the beam mesh? Table 8. This table is almost identical to Table 2. Remove and refer to Table 2. Complete Table 2 if some value is missed. Explain the difference between the tensile stress (center and ends). Line 380. All freedom degrees were fixed at the beam ends. How representative is this approach regarding the rotations? When L/d is too high, fixed rotations are not easy to perform. Could you please add any deatail about the physical fixation or reconsider these assumption?. Line 388. Complete the assumption of linearity. A linear solver implies the material linearity and small displacements. At the same time, the forces do not change their direction during the deformation. Figure 7. This figure the goal of this paper. This flowchart should show how the metodology should be applied by the readers.
Preliminary Modelling could be changed to "Preliminary Geometry Design". Loadings G,Q,W could depend on the used standard. Then, "Loads" could unifies them. In this block, "Boundary Conditions" should be included because they should be defined at this point. FEM Analysis and Structural Analysis should be unified in only one block because the means the same. The Effect of Changing Section Properties and dependent blocks should be moved to the end of flowchart, as proposals if the Preliminary Modelling is not able to support the Loads. Then, a desition-block Yes/No should be included with the question, are SLS, ULS and buckling under criteria? If Yes, finish. If No, an arrow should lead to CAD mofication through Thickness and / or Diameter change. Line 407. Describe the meaning of each variable after each equation. Some of them are described in following paragraphs, but not close. Table 9. Use GPa for the Elastic Modulus (E) as previus tables. Line 454. Use a reference for Timoshenko's Theory of Elastic Stability. Line 469. Where are the Appendixes 2-7? Line 470 - Line 474. Some spaces. Line 489. Rewrite "show" as "shows" Table 10. It shows the SLS:G+0.7Q+W (title), but SLS: G+Q and SLS:G+0.7Q+W have been included (first column). Fix this issue. it is recommended to split both cases. Table 10. Remove (5% deflection from the title). Line 491. How did you assessed the 5% of deflection? In which structure?
Line 500. Do not use italics in "the". Another interesting result to evaluate the efficiency is the ratio displacement-to-total lenght of used bamboo. It could highlight the more efficient way to use the material. In these results the Coordinate System should be included. DY means displacements along Y, but what is the Y-positive direction? Line 601. Change the title of this point, since the "material properties" has not been changed. Use "thickness and diameter" or "section properties".

I hope the authors find these comments useful and constructive.

Author Response

Reviewer’s Comments

Authors’ Reply

Try to reformulate the abstract, making it more concise and clear, describing the points sequentally. See the first specific comments

The required comment was addressed and changed.

Try to focus the paper on the method to design bamboo structures.The structure of the paper must be improved. The main objective is to develop a guidelines to design bamboo structures. Try to focus the paper in this direction. After finish the "Background review", the methodology should be described in detail.

The required comment was addressed and changed.

The main contribution of the current research is to suggest a simple and comprehensive numerical method to design a complicated curve structures which are extracted from bamboo. Since there is no a sound and reliable practical guideline to design curved shape structures made from bamboo, one of the main objectives is to address design issue in the grid structures. The indicated research methodology is theoretical analysis which is a combination of the analytical and numerical analysis. The relevant modelling strategy which is including the allocated section properties, material properties as well as the relevant boundary conditions is clearly indicated. The effect of the different joints on the different moment distribution is a separate research and it needs to allocate the different attention. 

Other points such as "3. Numerical Modelling Strategy", "4. The Applied Load" and "5. Numerical Analysis" should be described into the Methodology. The numerical modelling strategy and the numerical analysis could be unified to describe the CAD models (Designs 1 to 3) and the Finite Elements Analysis details.

The required comment was addressed and changed.

The point "7. The Preliminary Desgin Guidelines for bamboo" could be included in the conclusion point, once the guidelines is properly described in the methodology. The Figure 7 is your proposed method. It should be included and described as one of the most highligted content of your Metodology point. It is recommended to use passive perfect tenses instead present. See line 351 as an example ("has been constructed" instead "are constructed").

The required comment was addressed and changed.

Line 20. Do not use capital letter in "Bamboo".

The required comment was addressed and changed.

Line 24. In the abstract, the sentence "To address the issue, ... support system" should be moved after the sentence "These guidelines ... flexural situations". The main objective of this paper is to develop the design guidelines, while the finite element analylsis is a tool to aid the design.

The required comment was addressed and changed.

Line 26. The redundant sentence "Bamboo is known as ... than timber" could be combined with the first sentences in the abstract.

The required comment was addressed and changed.

Line 59. Remove "all three preliminary". Your method should be applicable to "any" design, not only for the three selected preliminary designs.

Of course, the current method can be applied to any cases; however, the three cases are presented to testify some particular design cases.

Line 60. Software AutoCad and Strand7 should include a reference.

The required comment was addressed and changed.

Line 64. Do not use capital letter in "Bamboo".

The required comment was addressed and changed.

Line 69. Remove the contraction in "bamboo's use".

The required comment was addressed and changed.

Line 71. Remove the extra point after the reference.

The required comment was addressed and changed.

Line 89. Rewrite the inconsisten sentence "Bamboo selected as the main ..." Table 1. Verify and contrast the range of Poisson's Ratio (up to 0.52 > 0.5) from the reference. Use another reference if necessary. Table 1. Try to unify the length units (mm, cm)

The required comment was addressed and changed.

Line 105. Move "From" to the next line.

The required comment was addressed and changed.

Line 138. Move "T" to the next line.

The required comment was addressed and changed.

Line 149. Remove the apostrophe.

The required comment was addressed and changed.

Line 175. Put "however" at the beginning of the sentence or rewrite the sentence, as more formal style.

The required comment was addressed and changed.

Line 184. At the end of the sentence "and bolted connections, with two options." What are the two options? If it refers to binding and bolted, try to reformulate the sentence.

The required comment was addressed and changed.

Line 193. Missing reference of Eurocode. (Line 193, Line 194, ...). Figure 1. Include the diameter dimension as drawing standars. Include the diamenter symbol and the diametral arrow.

The required comment was addressed and changed.

Line 223. Use "input" instead "inputted".

The required comment was addressed and changed.

Line 224. References could be merged as [25-27].

The required comment was addressed and changed.

Line 225. Rewrite "These values ... into Strand7" as "These values were applied to the beam elements in Strand7" Table 3. Use GPa as unit for Elastic Modulus (E) as Table 1. Table 3. Use the same Diameter (D) units as Table 1. Table 3. Use the same Thickness (t) units as Table 1.

All of the units were converted to MPa.

Line 310. The dead weight of the glass is a "weight". Why it is applied as pressure? Pressure acts in a normal direction, while weight "follows" the gravity direction.

In order to create possibility to be combined with the other types of the loading ( e.g Wind Loading), the gravity converted to the UDL or pressure.

Line 311. The dead weight of the bamboo is described as a density. Use daN instead kg. Line 315. Standars AS1170.1 and Eurocode 1 should be referenced.

The density will be converted to the weight per unit of meter. The relevant references were added.

Line 323. Standars AS1170.0 should be referenced.

The required comment was addressed and changed.

Line 333. Figure 8 appears before Figure 7 and Figure 6 in the text. Check if it refers to the Figure 7 in page 13. Move this figure frontwards if so. Figures should appear in the text in a sequential order.

It was checked, the order are correct.

Line 349 - Line 350. Live, dead and wind loads combinations have been abbreviated as G, Q, W. It could not be clear for a non-expert reader. Include the meaning of these abbreviations. Line 363. What is the element size (i.e. length) for the beam mesh? Table 8. This table is almost identical to Table 2. Remove and refer to Table 2. Complete Table 2 if some value is missed. Explain the difference between the tensile stress (center and ends)

The required comment was addressed and changed.

Line 380. All freedom degrees were fixed at the beam ends. How representative is this approach regarding the rotations? When L/d is too high, fixed rotations are not easy to perform. Could you please add any deatail about the physical fixation or reconsider these assumption?.

In current modelling the indicated technique was used in order to obtain a convergence solution.

Line 388. Complete the assumption of linearity. A linear solver implies the material linearity and small displacements. At the same time, the forces do not change their direction during the deformation. Figure 7. This figure the goal of this paper. This flowchart should show how the metodology should be applied by the readers.

I cannot follow this comment.

Preliminary Modelling could be changed to "Preliminary Geometry Design". Loadings G,Q,W could depend on the used standard. Then, "Loads" could unifies them. In this block, "Boundary Conditions" should be included because they should be defined at this point. FEM Analysis and Structural Analysis should be unified in only one block because the means the same. The Effect of Changing Section Properties and dependent blocks should be moved to the end of flowchart, as proposals if the Preliminary Modelling is not able to support the Loads. Then, a desition-block Yes/No should be included with the question, are SLS, ULS and buckling under criteria? If Yes, finish. If No, an arrow should lead to CAD mofication through Thickness and / or Diameter change.

All of the modelling part including the different analysis (including the buckling analysis ) are clearly explained .

There are number of valuable studies in the grid/bamboo structures, however, the current study is one of the pioneer studies using numerical methods to design curved shape structures which are made by bamboo. However, the suggested article was added to the current study.

Line 407. Describe the meaning of each variable after each equation. Some of them are described in following paragraphs, but not close. Table 9. Use GPa for the Elastic Modulus (E) as previus tables

All of the parameters are clearly described.

. Line 454. Use a reference for Timoshenko's Theory of Elastic Stability.

The relevant reference was added.

Line 469. Where are the Appendixes 2-7?

It was removed, as it was a mistake.

Line 470 - Line 474. Some spaces.

It was fixed.

Line 489. Rewrite "show" as "shows" Table 10. It shows the SLS:G+0.7Q+W (title), but SLS: G+Q and SLS:G+0.7Q+W have been included (first column). Fix this issue. it is recommended to split both cases. Table 10. Remove (5% deflection from the title).

It was fixed. The (5% deflection from the title), is relevant to the rate of the exaggeration in the deformed structures. 

Line 491. How did you assessed the 5% of deflection? In which structure?

The (5% deflection from the title), is relevant to the rate of the exaggeration in the deformed structures. 

Line 500. Do not use italics in "the". Another interesting result to evaluate the efficiency is the ratio displacement-to-total lenght of used bamboo. It could highlight the more efficient way to use the material. In these results the Coordinate System should be included. DY means displacements along Y, but what is the Y-positive direction?

The required comment was addressed and changed. A

Coordinate System was added.

Line 601. Change the title of this point, since the "material properties" has not been changed. Use "thickness and diameter" or "section properties".

The required comment was addressed and changed.

Reviewer 2 Report

Revision

Development of Preliminary Curved Bamboo Member Design Guidelines Through Finite Element Analysis

The paper needs of a major revision. It is too long and presents a lot of mistake (in paragraph number, figures, ref. etc..). The focus of the author is not evident. The Authors must substantially review the paper in order to have a more synthetic analysis and a more effective discussion of the results. 

The paper presents now the following structure:

Introduction Background review Numerical Modelling Strategy The Applied Load Numerical Analysis Results The effect of the material properties on the structural responses The Preliminary Design Guidelines for bamboo Conclusions

There are a lot of paragraph with many repetitions in the text. A summary is recommended. 

It is proposed the following structure:

Introduction State of the Art Numerical Analysis Discussion of results Conclusions

The suggested Title is (no clear indication about “Design Guidelines):

“Preliminary Curved Bamboo Member: Parametric Analysis and Design Recommendation”

The following questions have to find a proper response in the paper:

What is the focus of the author? What is the method or the “Modelling strategy” used? What is the innovative contribution of the Authors in the design process? What is the role of the joints in the analytic model? Are furnished clear and quantitative indication about Design?

These points need more attention:

Lines 177 – 184 - 2.5. Connection Type

The role of the connection between member is very relevant for this structural member. The analysis of the following reference is recommended:

Froli, M.; Mariani, G.; Ngoma, I.; Sassu, M. A pilot test on the problem of joining steel plates to bamboo rods. In Proceedings of the Department of Structural Engineering—University of Pisa, Pisa, Italy, 2003.

Sassu, M.; De Falco, A.; Giresini, L.; Puppio, M.L. Structural Solutions for Low-Cost Bamboo Frames: Experimental Tests and Constructive Assessments. Materials 2016, 9, 346.

Lines 146 – 154 - 2.3 Grid Shell

The approach to the design of Grid shell Structures is too simplistic. An in-depth study is appropriate:

Corio E., Laccone F., Pietroni N., Cignoni P., Froli M. (2017). Conception And Parametric Design Workflow For A Timber Large-Spanned Reversible Grid Shell To Shelter The Archaeological Site Of The Roman Shipwrecks In Pisa, in International Journal of Computational Methods and Experimental Measurements, WIT Press, Vol. 5 No. 4, pp. 551-561; DOI: 10.2495/CMEM-V0-N0-1-11

Froli M., Laccone F. (2017). Experimental static and dynamic tests on a large-scale free-form Voronoi grid shell mock-up in comparison with finite-element method results, in International Journal of Advanced Structural Engineering, Springer; DOI: 10.1007/s40091-017-0166-9

Lines 722-728  - “there is no confirmation of true validity” – ???

Not Eligible! What is the contribution of the Authors to Scientific Progress?

In addition, the evaluation of the Authors about the mechanical properties of the bamboo beams or  trusses are not eligible. Scientific literature presents a lot of study about bamboo structure. See for example:

Sharma, B.; Harries, K.; Ghavami, K. Methods of determining transverse mechanical properties of full-culm bamboo. Constr. Build. Mater. 2013, 38, 627–637.

Author Response

The second reviewer’s comments

The reviewer’s comments

Authors’ reply

Introduction Background review Numerical Modelling Strategy The Applied Load Numerical Analysis Results The effect of the material properties on the structural responses The Preliminary Design Guidelines for bamboo Conclusions

There are a lot of paragraph with many repetitions in the text. A summary is recommended. 

Please consider the following reply.

What is the focus of the author? What is the method or the “Modelling strategy” used? What is the innovative contribution of the Authors in the design process? What is the role of the joints in the analytic model? Are furnished clear and quantitative indication about Design?

These points need more attention:

The main contribution of the current research is to suggest a simple and comprehensive numerical method to design a complicated curve structures which are extracted from bamboo. Since there is no a sound and reliable practical guideline to design curved shape structures made from bamboo, one of the main objectives is to address design issue in the grid structures. The indicated research methodology is theoretical analysis which is a combination of the analytical and numerical analysis. The relevant modelling strategy which is including the allocated section properties, material properties as well as the relevant boundary conditions is clearly indicated. The effect of the different joints on the different moment distribution is a separate research and it needs to allocate the different attention.  

Lines 177 – 184 - 2.5. Connection Type

The role of the connection between member is very relevant for this structural member. The analysis of the following reference is recommended:

Froli, M.; Mariani, G.; Ngoma, I.; Sassu, M. A pilot test on the problem of joining steel plates to bamboo rods. In Proceedings of the Department of Structural Engineering—University of Pisa, Pisa, Italy, 2003.

Sassu, M.; De Falco, A.; Giresini, L.; Puppio, M.L. Structural Solutions for Low-Cost Bamboo Frames: Experimental Tests and Constructive Assessments. Materials 2016, 9, 346.

The suggested articles were added to the current study.

Lines 146 – 154 - 2.3 Grid Shell

The approach to the design of Grid shell Structures is too simplistic. An in-depth study is appropriate:

Corio E., Laccone F., Pietroni N., Cignoni P., Froli M. (2017). Conception And Parametric Design Workflow For A Timber Large-Spanned Reversible Grid Shell To Shelter The Archaeological Site Of The Roman Shipwrecks In Pisa, in International Journal of Computational Methods and Experimental Measurements, WIT Press, Vol. 5 No. 4, pp. 551-561; DOI: 10.2495/CMEM-V0-N0-1-11

Froli M., Laccone F. (2017). Experimental static and dynamic tests on a large-scale free-form Voronoi grid shell mock-up in comparison with finite-element method results, in International Journal of Advanced Structural Engineering, Springer; DOI: 10.1007/s40091-017-0166-9

The suggested articles were added to the current study.

Lines 722-728  - “there is no confirmation of true validity” – ???

Not Eligible! What is the contribution of the Authors to Scientific Progress?

Since there is no a sound and reliable practical guideline to design curved shape structures made from bamboo, one of the main objectives is to address design issue in the grid structures. The indicated research methodology is theoretical analysis which is a combination of the analytical and numerical analysis. The relevant modelling strategy which is including the allocated section properties, material properties as well as the relevant boundary conditions is clearly indicated.

In addition, the evaluation of the Authors about the mechanical properties of the bamboo beams or  trusses are not eligible. Scientific literature presents a lot of study about bamboo structure. See for example:

Sharma, B.; Harries, K.; Ghavami, K. Methods of determining transverse mechanical properties of full-culm bamboo. Constr. Build. Mater. 2013, 38, 627–637

There are number of valuable studies in the grid/bamboo structures, however, the current study is one of the pioneer studies using numerical methods to design curved shape structures which are made by bamboo. However, the suggested article was added to the current study.

Reviewer 3 Report

The authors present a paper on the use of curved bamboo. The work presented is interesting and presents some level of innovation. In general the theme is properly framed, the experimental plan well designed, the results duly presented with conclusions supported by them.

Overall I am of the opinion that the paper has conditions for publication. However, I have only 2 comments that may in some way contribute to improving the overall quality of the document.

Regarding the abstract, it is essential that, in addition to scope, motivation, methodology and results, the authors highlight the expected impact of the research.

Results: In general this chapter lacks the intersection between the results obtained and their comparison with values of other works and authors. Benchmarketing is essential for this type of work.

Author Response

The Third reviewer’s comments

The reviewer’s comments

Authors’ reply

Regarding the abstract, it is essential that, in addition to scope, motivation, methodology and results, the authors highlight the expected impact of the research.

The main contribution of the current research is to suggest a simple and comprehensive numerical method to design a complicated curve structures which are extracted from bamboo. Since there is no a sound and reliable practical guideline to design curved shape structures made from bamboo, one of the main objectives is to address design issue in the grid structures. The indicated research methodology is theoretical analysis which is a combination of the analytical and numerical analysis. The relevant modelling strategy which is including the allocated section properties, material properties as well as the relevant boundary conditions is clearly indicated. The effect of the different joints on the different moment distribution is a separate research and it needs to allocate the different attention. 

Results: In general this chapter lacks the intersection between the results obtained and their comparison with values of other works and authors. Benchmarketing is essential for this type of work.

There are number of valuable studies in the grid/bamboo structures, however, the current study is one of the pioneer studies using numerical methods to design curved shape structures which are made by bamboo. However, the suggested article was added to the current study. Thus, the current study can be considered a significant numerical benchmark by to date.

Round 2

Reviewer 2 Report

The work can be accepted in the present form.
Kind regards.